# Process Simulation of Steam Gasification of Torrefied Woodchips in a Bubbling Fluidized Bed Reactor Using Aspen Plus

Nhut M. Nguyen [1,2,*], Falah Alobaid [1] and Bernd Epple [1]

[1] Institute for Energy Systems and Technology, Technical University of Darmstadt, Otto-Berndt-Straße 2, 64287 Darmstadt, Germany; falah.alobaid@est.tu-darmstadt.de (F.A.); bernd.epple@est.tu-darmstadt.de (B.E.)

[2] Department of Chemical Engineering, Campus II 3/2 Street, Can Tho University, Can Tho 900000, Vietnam

* Correspondence: nhut.nguyen@est.tu-darmstadt.de; Tel.: +49-6151-16-22673

**Abstract:** A comprehensive process model is proposed to simulate the steam gasification of biomass in a bubbling fluidized bed reactor using the Aspen Plus simulator. The reactor models are implemented using external FORTRAN codes for hydrodynamic and reaction kinetic calculations. Governing hydrodynamic equations and kinetic reaction rates for char gasification and water-gas shift reactions are obtained from experimental investigations and the literature. Experimental results at different operating conditions from steam gasification of torrefied biomass in a pilot-scale gasifier are used to validate the process model. Gasification temperature and steam-to-biomass ratio promote hydrogen production and improve process efficiencies. The steam-to-biomass ratio is directly proportional to an increase in the content of hydrogen and carbon monoxide, while gas yield and carbon conversion efficiency enhance significantly with increasing temperature. The model predictions are in good agreement with experimental data. The mean error of $CO_2$ shows the highest value of 0.329 for the steam-to-biomass ratio and the lowest deviation is at 0.033 of carbon conversion efficiency, respectively. The validated model is capable of simulating biomass gasification under various operating conditions.

**Keywords:** steam gasification; biomass; bubbling fluidized bed; Aspen Plus simulation; hydrogen production





## 1. Introduction

Biomass has been considered as one of the most important primary and renewable energy resources for the production of heat, electricity, hydrogen, chemicals, and liquid fuels due to its carbon-neutral renewable and abundant quantity. Furthermore, the energy production from biomass is advantageous to other renewable sources such as wind energy, hydropower, solar energy, etc. [1].

Gasification is a partial oxidation process at high temperatures, which can convert organic or fossil fuel-based carbonaceous materials into gaseous fuel including mainly $H_2$, CO, $CO_2$, and $CH_4$. In the presence of steam, the product gas has been generated with 30–60% of $H_2$ content and a calorific value of 10–18 MJ/Nm$^3$ [2,3]. Thus, steam gasification of biomass is an effective and efficient technology for sustainable hydrogen production without a carbon footprint [4].

Generally, three reactor configurations can be used for biomass gasification, i.e., the entrained flow, fixed bed, and fluidized bed reactors. The fluidized bed gasifiers show advantages for biomass conversion due to the perfect contact between gas and solid, increasing heat and mass transfer characteristics, and improving temperature control. An experimental investigation of gasification of torrefied woody biomass was conducted by Berrueco et al. [5] in a pressurized fluidized bed, evaluating the effect of pressure and torrefaction level on the yield and composition of the products. The authors found that the

pressure could result in a decline in CO and $H_2$ levels, whereas $CO_2$ and $CH_4$ yield increase. Chen et al. [6] observed that the cold gasification efficiency of torrefied bamboo rose by 88% compared to raw bamboo. Furthermore, char gasification reactions are one of the most important reactions in biomass gasification. Many kinetic studies have been carried out through the thermogravimetric analysis instrument (TGA) to determine the kinetic parameters of char gasification [7–12]. Some reaction kinetic models have been proposed for char gasification. They could be categorized into two groups such as theoretical and semiempirical models. Four conversion models have been investigated for char gasification reactions, i.e., volumetric model, shrinking core model, random pore model, and Johnson model [9,13]. Additionally, the Langmuir-Hinshelwood reaction model has been used extensively as a kinetic model for heterogeneous reactions, particularly char gasification with steam and carbon dioxide [14].

Along with the computational progress, the numerical simulation could help bypass a long planning and construction process of experimental studies, provide low-cost methods for the proper design and project realization. Mathematical models are developed to describe the physical and chemical phenomena occurring inside the gasifier and to understand the effect of various operating and design parameters on the process performance. The model also is used to predict the behavior of gasification at off-design conditions and the optimum operating parameters [15]. The main simulation methods can be categorized as the thermal equilibrium model, kinetic model, numerical model, and artificial neural network [15–27]. The thermodynamic equilibrium model is simple and provides the preliminary comparison and assessment of the gasification process [28]. There are three equilibrium modeling approaches, such as the restricted equilibrium model, empirical correlation-based model, and the model based on a combination of hydrodynamic and kinetic aspects [29].

The Aspen Plus process simulator, which is developed to facilitate physical, chemical, and biological calculations, has been used commonly in various studies to simulate coal and biomass gasification. Due to the complex nature of tar, most of the studies in the literature have not considered tar calculations. Tar is a product of the thermal decomposition process of biomass, including condensed oils such as olefins, phenols, aromatics, etc. Few Aspen Plus models of biomass gasification have been reported on tar and its kinetics. For example, a mathematical model of biomass gasification in a bubbling bed reactor was developed in Aspen Plus with a sub-model for tar generation and cracking [30]. This study has defined tar and its cracking kinetics to improve the model performance and its credibility. Nikoo et al. [31] proposed a process model for biomass gasification using external FORTRAN subroutines for both hydrodynamic and reaction kinetic calculations simultaneously in Aspen Plus. A process model was developed to simulate the air-stream gasification of biomass in a bubbling fluidized bed reactor [17]. This model was based on chemical reaction rates, empirical correlations of pyrolysis mass yields, and hydrodynamic parameters. A simulation was performed in the Aspen program for steam gasification of rice husk to evaluate the influence of gasification temperature and steam-to-biomass ratio on product gas composition [32]. The model was developed based on the chemical equilibrium to predict the gas composition of the process. A model is developed based on Gibbs free energy minimization applying the restricted equilibrium method to study the influence of key parameters on the performance of steam gasification of biomass [33].

The high amount of volatile matter in biomass and the complexity of biomass reaction kinetics in fluidized beds have hindered the simulation of biomass gasification. Many studies ignored the kinetics of char gasification and developed their gasification model based on Gibbs equilibrium. Additionally, in a typical fluidized bed gasifier, solid fuels and bed material are fluidized by a mixture of gases resulting in good solid-gas heat and mass transfer. Consequently, hydrodynamic behavior is a crucial factor in a fluidized bed gasifier influencing strongly the performance of the gasification process. Therefore, reliable process simulation studies on biomass gasification are still limited, resulting in a lack of understanding of the fundamentals of the biomass-based gasification process.

This study has been developed in Aspen Plus based on the previous studies with some improvements to provide a good understanding of biomass gasification in terms of the effect of operating parameters on the process performance and the phenomena occurring in a bubbling fluidized bed gasifier. The proposed model based on a combination of both hydrodynamic and reaction kinetic calculations simultaneously is capable of predicting the steady-state performance of a negative gauge pressure bubbling fluidized bed gasifier. Due to the complexity of biomass characteristics, the mass yields of pyrolytic products released from torrefied wood chips were determined by the model of Neves et al. [34] according to the biomass proximate and ultimate analyses. Char gasification kinetics obtained from experimental investigations are to determine the reaction rate of char gasification. Three chemical reactions, i.e., char gasification with steam and $CO_2$ and water–gas shift reactions are taken into consideration in the model to calculate variations of components in biomass gasification. Due to the lack of a library model to simulate fluidized bed units in the Aspen Plus simulator, external FORTRAN codes are implemented with input data to simulate an operation of a bubbling fluidized bed gasifier. The validity and accuracy of the model are evaluated by comparing the numerical result obtained with the experimental data of biomass steam gasification.

## 2. Modeling Methods

In the test rig, silica sand is used as bed material. Biomass is fed continuously and reacts with steam to produce syngas, mainly comprising hydrogen, carbon monoxide, carbon dioxide, and methane. The model is developed based on hydrodynamic and reaction rate kinetic calculations at the isothermal condition.

### 2.1. Process Assumption

For the modeling of the biomass gasification process, the following assumptions were considered:

- The process is modeled in steady-state and isothermal conditions.
- The reactive gases are $H_2$, CO, $CO_2$, $CH_4$, and $H_2O$.
- $N_2$, $NH_3$, $H_2S$, and $SO_2$ are considered chemically inert components in gasification reactions.
- The char is modeled with only components of carbon black and ash.
- All gases are uniformly distributed within the emulsion phase.
- Particles are spherical and of uniform size. Their average diameter remains unchanged during the gasification.
- Char gasification starts in the dense zone and completes in the freeboard.
- Ash and sand are chemically inert under process conditions.

For the hydrodynamic calculation, the following assumptions were made:

- There are two regions in the fluidized bed reaction: bed and freeboard.
- The bubbling regime is maintained in the bed region.
- The volumetric flow rate of gas increases along with the reactor height, corresponding to the gas products generated.
- The mixing of solid particles in the reactor is perfect.
- The reactor is divided into many control volumes with constant hydrodynamic parameters.
- The fluidized bed is one-dimensional.

### 2.2. Experimental Facility

The experimental reactor comprises a circular column with 54.5 mm inner diameter and 550 mm length, and a porous gas distributor plate at the bottom shown in Figure 1. Pressure and temperature sensors are installed at 90, 350, and 550 mm along the reactor. Two electrical heating elements are used to heat the reactor. 800 g of silica sand is filled up in the reactor as bed material due to its good mechanical properties and no active role. Sand's properties are shown in Table 1. The solid fuel is filled in a hopper and fed continuously into the reactor at 90 mm height through a screw feeder. A gas mixture is pre-heated to 300 °C before being injected into the reactor through a porous distributor. A

part of the product gas is extracted from the reactor to a gas analysis unit, ABB URAS 206 analyzers. A summary of the continuous measuring methods and the maximum relative error is shown in Table 2.

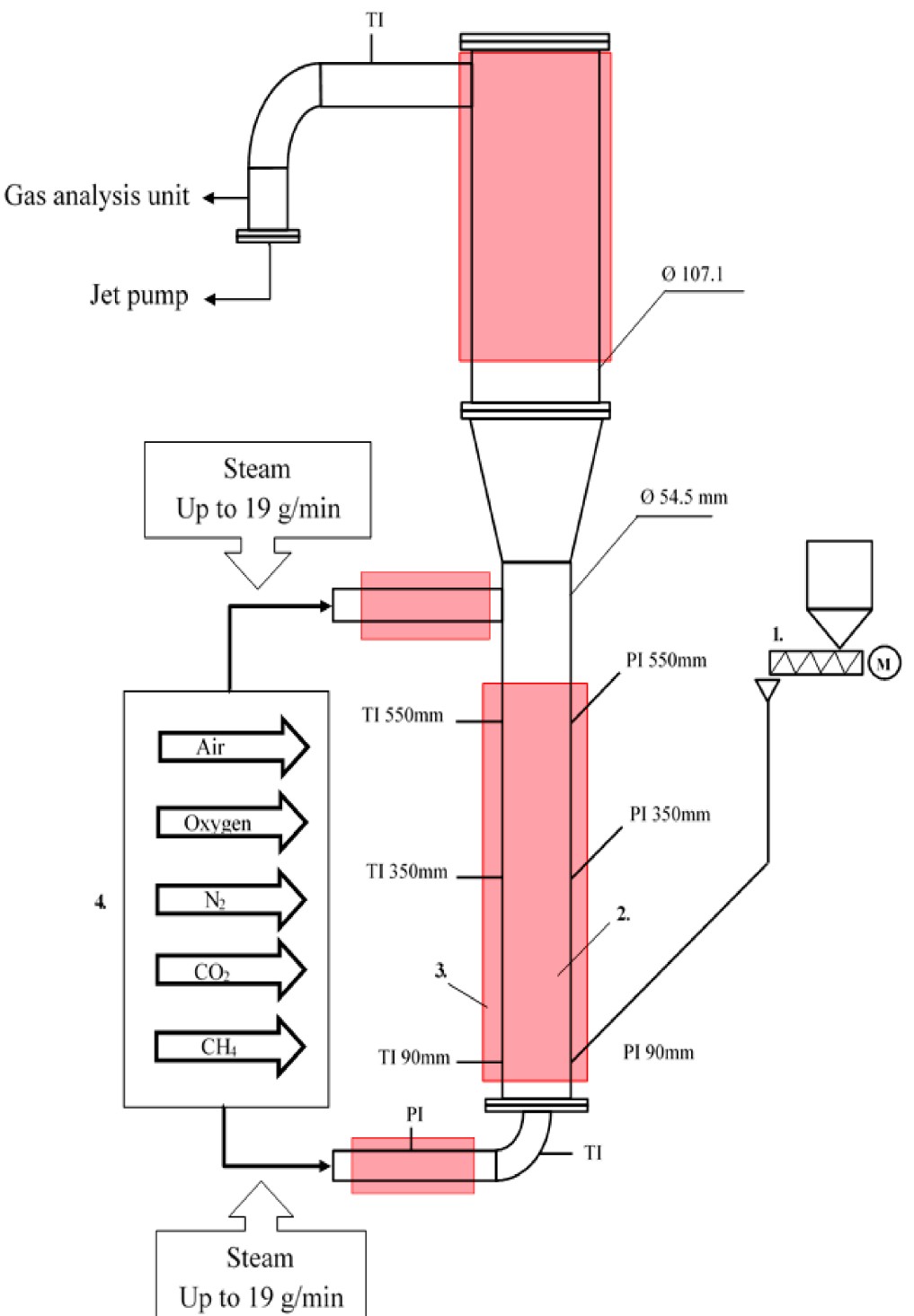

**Figure 1.** Schematic configuration of the bubbling fluidized bed test rig. 1—screw conveyor for feeding fuel; 2—bubbling fluidized bed reactor; 3—electrical heater; 4—gas distribution system.

**Table 1.** Experimental setup parameters used in the simulation.

| Fluidized Bed Reactor | |
|---|---|
| Temperature (°C) | 800–900 |
| Pressure (bar) | 0.92 |
| Diameter (m) | 0.0545 |
| Height (m) | 0.55 |
| Bed Material (Silica Sand) | |
| Mean particle size (m) | $177 \times 10^{-6}$ |
| Density (kg/m$^3$) | 2650 |
| Mass weight (kg) | 0.8 |
| Steam | |
| Temperature (°C) | 300 |
| Flow rate (kg/h) | 0–0.084 |
| Concentration (%) | 0–33.33 |

**Table 2.** The method for continuous measurement of the gas composition.

| Species | Method | Range | Unit | Rel. Error in % |
|---|---|---|---|---|
| $CO_2$ | Infrared | 0–100 | Vol.% | <0.5 |
| CO | Infrared | 0–20 | Vol.% | <0.5 |
| $CH_4$ | Infrared | 0–5 | Vol.% | <0.5 |
| $H_2$ | Paramagnetic | 0–20 | Vol.% | <0.5 |
| $O_2$ | Paramagnetic | 0–25 | Vol.% | <0.5 |

*2.3. Reaction Kinetics*

The biomass gasification in the presence of steam includes a series of complex and competing reactions, including homogeneous and heterogeneous reactions. These reactions occur simultaneously in four overlapping steps, i.e., drying, thermal decomposition, oxidation, and reduction. The overall reaction of biomass steam gasification is described as follows:

$$\text{Biomass} + H_2O \rightarrow H_2 + CO + CO_2 + CH_4 + l\text{ hydrocarbon} + \text{Tar} + \text{Char} \qquad (1)$$

Initially, the moisture content of biomass reduces to less than 5% [35]. Then, devolatilization and cracking of weaker chemical bonds takes place at a temperature ranging from 250 to 700 °C [36], producing various fractions: gas, a liquid/condensed, and a solid [37–41]. In this step, the biomass converts into solid char which can range from 5% to 10% for fluidized bed reactors, or 20% to 25% for fixed-bed reactors [37–41]. The volatiles and solid char react with limited oxygen in the oxidation stage to produce mainly CO, $CO_2$, and $H_2O$, and the heat from this stage can supply to the endothermic reactions ((2) and (3)). The unreacted char is converted by steam and $CO_2$ to form the final gaseous products [36,42]. The yield of hydrogen from steam gasification is significantly higher than that of fast pyrolysis followed by a steam reforming of char.

$$C + \alpha H_2O \rightarrow (2 - \alpha)\,CO + (\alpha - 1)\,CO_2 + \alpha H_2 \text{ (Water-gas reaction)} \qquad (2)$$

$$C + CO_2 \rightarrow 2CO \text{ (Boudourd reaction)} \qquad (3)$$

where $\alpha$ has been experimentally determined in the range of 1.5–1.1 at 750–900 °C [43]. For the proposed model, the selected value of $\alpha$ was 1.3, showing good agreement with experimental data.

Water–gas shift reaction and steam reforming reaction occur simultaneously according to gasification conditions, playing a key factor for hydrogen production:

$$CO + H_2O \rightarrow CO_2 + H_2 \qquad (4)$$

$$CH_4 + H_2O \rightarrow CO + H_2 \tag{5}$$

The carbon in the char is gasified with steam and $CO_2$ ((2) and (3)). The reaction rate kinetics of char gasification are calculated when fluidized with steam and carbon dioxide as follows [44]:

$$\left[\frac{dX_C}{dt}\right]_{H_2O} = \frac{k_{H_2O}P_{H_2O}}{1 + K_{H_2O}P_{H_2O} + K_{H_2}P_{H_2}}(1 - X_C) \tag{6}$$

$$\left[\frac{dX_C}{dt}\right]_{CO_2} = \frac{k_{CO_2}P_{CO_2}}{1 + K_{CO_2}P_{H_2O} + K_{CO}P_{CO}}(1 - X_C) \tag{7}$$

The kinetic parameters of char gasification can be found in Table 3.

**Table 3.** Kinetic parameters for char gasification obtained from experimental results.

| $H_2O$ | | $CO_2$ | | Unit |
|---|---|---|---|---|
| $k_{0,H_2O}$ | $1.02 \times 10^{11}$ | $k_{0,CO_2}$ | $9.62 \times 10^{10}$ | $kPa^{-1}\ min^{-1}$ |
| $E_{a1,H_2O}$ | 281.86 | $E_{a1,CO_2}$ | 284.36 | $kJ/mol$ |
| $K_{0,H_2O}$ | 60.34 | $K_{0,CO_2}$ | 3.63 | $kPa^{-1}$ |
| $E_{a2,H_2O}$ | 61.69 | $E_{a2,CO_2}$ | 40.08 | $kJ/mol$ |
| $K_{0,H_2}$ | $1.56 \times 10^{-10}$ | $K_{0,CO}$ | $2.24 \times 10^{-10}$ | $kPa^{-1}$ |
| $E_{a3,H_2}$ | $-203.46$ | $E_{a3,CO}$ | $-195.64$ | $kJ/mol$ |

The water–gas shift reaction (WGSR) (R4) takes place in a homogeneous phase. In this model, the WGSR is assumed to occur in all regions in the reactor. Thus, the reaction rate kinetics of the WGSR is calculated as [45]:

$$\frac{dF_{WGS,i}}{dV_j} = (1 - \varepsilon_j)k_{0,WGS}e^{-\frac{E_{WGS}}{RT}}C_{CO}^{0.5}C_{H_2O} \tag{8}$$

where $\varepsilon_j$ is the porosity of the bed in the region $j$ with volume $V_j$. The pre-exponential factor of the kinetic constant is $k_{0,WGS} = 7.97 \times 10^9\ (m^3/mol)^{0.5}.s^{-1}$ and the activation energy is $E_{WGS} = 274.5\ kJ/mol$.

### 2.4. Hydrodynamic Calculation

The hydrodynamic properties of the bubbling fluidized bed reactor have a significant effect on the fuel conversion during biomass gasification. Here, the calculation equations and empirical correlations, reported in the literature, have been used to determine the hydrodynamic parameters, considering that the model is divided into two regions: bed and freeboard.

#### 2.4.1. Bed Hydrodynamic

The minimum fluidization velocity for small particles have been introduced by Kunii and Levenspil [46] as follows:

$$Ar = \frac{d_p^3 \rho_g (\rho_s - \rho_g) g}{\mu^2} \tag{9}$$

$$u_{mf} = \frac{33.7\mu}{d_p\rho_g}\left(\sqrt{1 + 3.59 \times 10^{-5}Ar} - 1\right) \tag{10}$$

The correlations were developed to determine the volume fraction occupied by bubbles in a fluidized bed [32].

$$B = 1 + \frac{10.978\left(u_f - u_{mf}\right)^{0.738} \rho_s^{0.376} d_p^{1.006}}{u_{mf}^{0.937} \rho_g^{0.126}} \tag{11}$$

$$\delta_b = 1 - 1/B \tag{12}$$

The following relation gives the bed void fraction:

$$\varepsilon_f = \delta_b + (1 - \delta_b)\varepsilon_{mf} \tag{13}$$

### 2.4.2. Freeboard Dynamics

The volume fraction of solid varies along with the height of the freeboard. The void fraction of the freeboard is determined by the following equation.

$$1 - \varepsilon_{fb} = (1 - \varepsilon_f)\exp(-az) \tag{14}$$

where a is the decay constant of solid particles in the freeboard, a is determined from the graph with the following range [47]:

$$a = \frac{1.3}{u_f} \tag{15}$$

with:

$$u_f \leq 1.25 \text{ m/s} \tag{16}$$

$$d_p \leq 800 \text{ μm} \tag{17}$$

### 2.5. Aspen Plus Model

The biomass gasification model involves various stages in Aspen Plus. The overall gasification process is illustrated in Figure 2. The biomass decomposition is simulated in the RYIELD block. The product distribution is determined by the model of Neves et al. [34] based on the proximate and ultimate analyses of biomass (listed in Tables 4 and 5). The volatile components obtained from pyrolysis simulated the volatile reactions in the RGIBBS reactor with the assumption that these reactions follow the Gibb equilibrium. The char gasification is modeled in two RSTOIC reactors, corresponding to bed and freeboard. The hydrodynamics and kinetics have been written in two external FORTRAN codes. The products then go through CYCLONE to separate gas products from solid impurities.

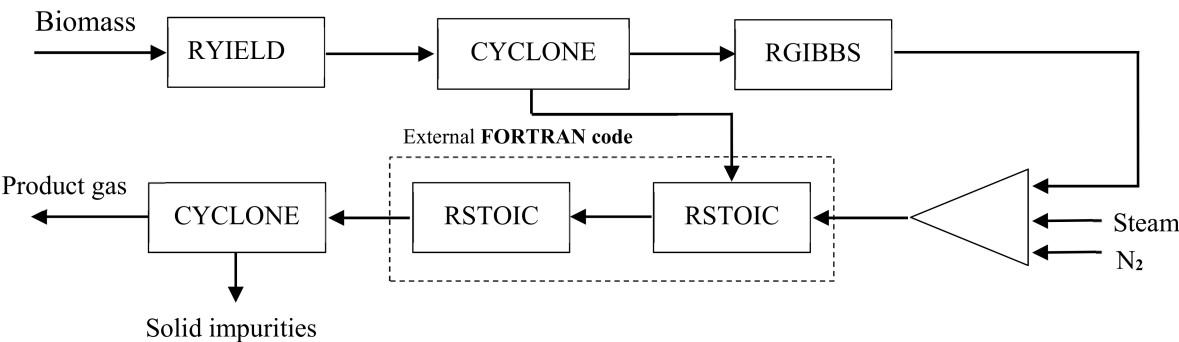

**Figure 2.** Biomass gasification model flow chart in Aspen Plus.

**Table 4.** Main properties of the torrefied woodchips.

|  | Property | Value | Note |
|---|---|---|---|
| Proximate analysis (wt.%) | Moisture | 5.28 | As received |
|  | Volatile matter | 70.75 | As received |
|  | Fixed carbon | 22.82 | As received |
|  | Ash | 1.15 | As received |
| Ultimate analysis (wt.% daf) | C | 54.46 | Dry basis |
|  | H | 5.99 | Dry basis |
|  | O | 39.31 | Dry basis |
|  | N | 0.24 | Dry basis |
|  | S | 0.00254 | Dry basis |
| HHV (MJ/kg) |  | 20.97 | As received |
| LHV (MJ/kg) |  | 19.26 | As received |
| Bulk density (kg/m$^3$) |  | 161.71 | As received |
| Mean particle diameter (μm) |  | 296.65 | Mass-weighted average diameter |

**Table 5.** Mass yield distribution (wt.%) from biomass decomposition.

| Component | Wt.% | Component | wt.% |
|---|---|---|---|
| Ash | 1.09 | $H_2O$ | 11.05 |
| CO | 46.9 | $N_2$ | 0.22 |
| C | 17.53 | $CO_2$ | 6.69 |
| $CH_4$ | 15.39 | $H_2S$ | 0.0025 |
| $H_2$ | 1.12 |  |  |

### 2.5.1. Biomass Characteristics

The torrefied woodchips, a feedstock in this study, were ground and sieved to a particle size of 200 to 850 μm. The proximate analysis of all samples was conducted following the DIN norms 18122, 18123, and 18134 standard test methods for ash, volatile matter, fixed carbon, and moisture determination, respectively. The ultimate analysis was carried out using an elemental analyzer (Elementar vario MACRO cube) with a measurement deviation <0.1%. The feedstock characteristics are shown in Table 4.

### 2.5.2. Biomass Decomposition

Devolatilization (pyrolysis) is a thermal decomposition of biomass at high temperatures in an inert atmosphere. Biomass pyrolysis includes extremely complex reactions that convert biomass into a mixture of gases, char, and liquid (tars). In this work, the Aspen Plus yield reactor, RYIELD, is used to simulate the decomposition of biomass. The mass yield distribution of pyrolysis products is derived from a pyrolysis model [34] based on the biomass proximate and ultimate analyses. Tars and larger hydrocarbons are assumed to be converted directly to methane and carbon monoxide in this study. The summary of mass product yields from the biomass decomposition is presented in Table 5.

### 2.5.3. Char Gasification

Char gasification is performed in the Aspen Plus STOIC reactor, RSTOIC, by using an external FORTRAN code to calculate hydrodynamic parameters and reaction rate kinetics. As mentioned above, the reactor is divided into two regions, bed and freeboard, each region is simulated by one RSTOIC. In FORTRAN code, each RSTOIC is divided into a finite number of equal volumes. Hydrodynamic parameters are determined by a series of equations and correlations in Section 2.4. Biomass gasification is a complex series of competing reactions. To simplify the process, this study only takes account of the water gas reaction, Boudourd reaction, and the water–gas shift reaction in the kinetic calculation. Their reaction rate kinetics are presented in Section 2.3.

### 2.5.4. Calculation Procedure

The model equations, given in previous sections, are implemented in Fortran codes. A flow chart for the calculation procedure is shown in Figure 3. Input data, such as reactor configuration, gasification conditions, the characteristics of biomass, bed material and gases, kinetic parameters, etc. are described in Tables 1–5. Firstly, it is necessary to make assumptions of molar values of components and initial carbon conversion in the reactor as well as bed properties such as bed height and bed volume. For the analysis, the volume of the reactor is divided into N divisions. Then, hydrodynamic and kinetic aspects are calculated discretely for dense and lean zones through the model equations described in previous sections, employing an iterative procedure. The calculation ends once all divisions have been calculated and the error does not exceed $10^{-4}$. The output data, such as the concentration profiles of components, solid conversion and fluidization properties, etc. are obtained along the entire reactor.

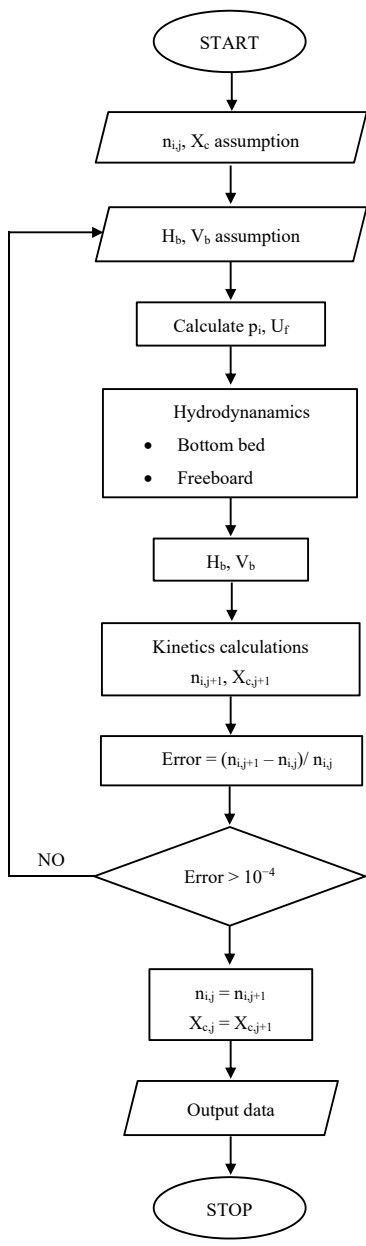

**Figure 3.** Simplified flow diagram of simulation calculation.

*2.6. Model Validation*

Simulation results were compared with experimental data and the deviation between simulation and experiment was determined. The sum of the squared deviation method is applied to determine the accuracy of the simulation.

$$RSS = \sum_{i=1}^{N} \left( \frac{y_{ie} - y_{ip}}{y_{ie}} \right)^2 \tag{18}$$

$$RSS = \frac{RSS}{N} \tag{19}$$

$$Mean\ error = \sqrt{MRSS} \tag{20}$$

## 3. Results and Discussion

This study investigated the effects of important parameters, namely, steam-to-biomass ratio (SBR) and gasification temperature during steam gasification of biomass through the Aspen Plus process flow model at steady-state conditions. One parameter varies, while the others are kept constant. To validate the simulation results, the experimental data from steam gasification of torrefied biomass in a bubbling fluidized bed reactor were used [47].

The isothermal experimental investigations of steam gasification of biomass were carried out in a pilot-scale bubbling fluidized bed reactor (Figure 1) using silica sand as bed material. The performance of biomass gasification was analyzed to assess the influence of operating parameters, i.e., gasification temperature, steam-to-biomass ratio, equivalence ratio. During the investigations, each operating parameter varied in the desired range, while other parameters were fixed at known values. After reaching steady conditions, the variations of gas composition were recorded and analyzed for the process performance. A detailed experimental description and its results are presented in a publication [47].

*3.1. Effect of Gasification Temperature*

The gasification temperature is a crucial parameter in biomass gasification. In this study, three temperatures ranging between 800 and 900 °C were investigated, while the steam-to-biomass ratio was fixed at 1.2 for the steam gasification. The effects of gasification temperature on the gas composition (on dry basic and $N_2$ free) are presented in Figure 4. The simulation results indicate that the content of $H_2$ increases with elevated temperature, while the $CH_4$ content decreases considerably. Furthermore, the content of CO rose from 800 to 850 °C, then dropped down to 21.98% at 900 °C. It is noteworthy that the $CO_2$ fraction showed an opposite trend. Moreover, operating temperature also enhanced the gas yield produced and CCE by 1.28 $Nm^3/kg_{biomass}$ and 51.97% from 800 to 900 °C, respectively (shown in Figures 5 and 6). A similar trend is observed in the experimental results.

The effect of operating temperature on the process performance of steam gasification of biomass can be attributed to endothermic and exothermic reactions in biomass gasification [42,48]. According to Le Chatelier's principle and dynamic equilibrium, the endothermic reactions ((2), (3), and (5)) are strengthened with elevated temperature, increasing the contents of hydrogen and carbon monoxide, while reducing in methane fraction. Additionally, the water–gas shift reaction can be promoted by the high contents of carbon monoxide and steam in the gasifier, resulting in a decline of carbon monoxide in the product gas. These reactions take place simultaneously in the reactor; thus, there is a considerable rise in $H_2$ content and fluctuations in the values of other components at high temperatures.

Elevated temperature favors the rate of char gasification reactions and the water–gas shift reaction, which can result in increased dry gas yield and the decreasing unreacted char. Therefore, increasing operating temperature improves biomass conversion and hydrogen production in steam gasification of torrefied biomass.

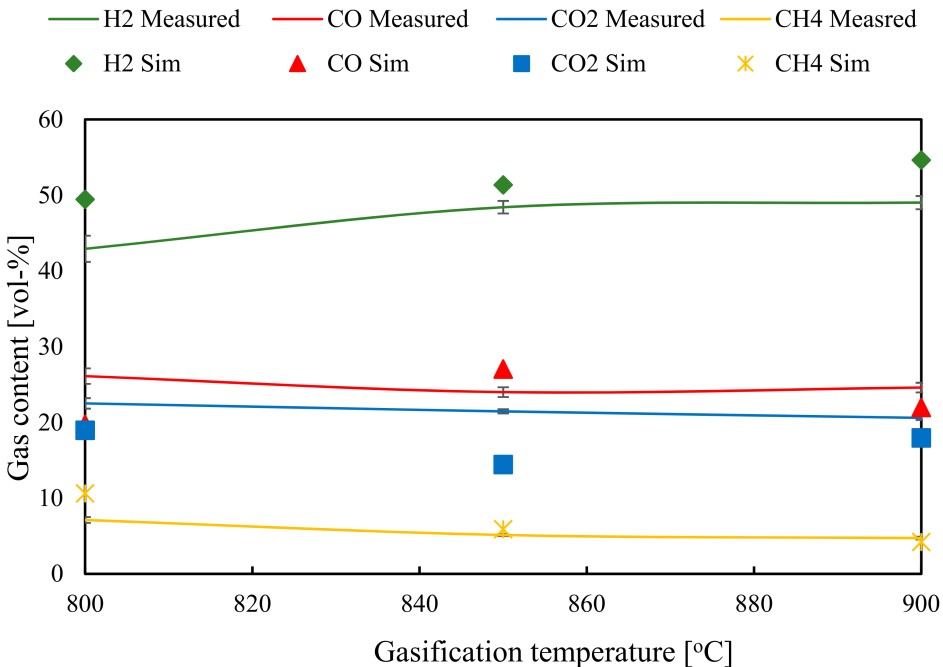

**Figure 4.** Effect of gasification temperature on the gas composition.

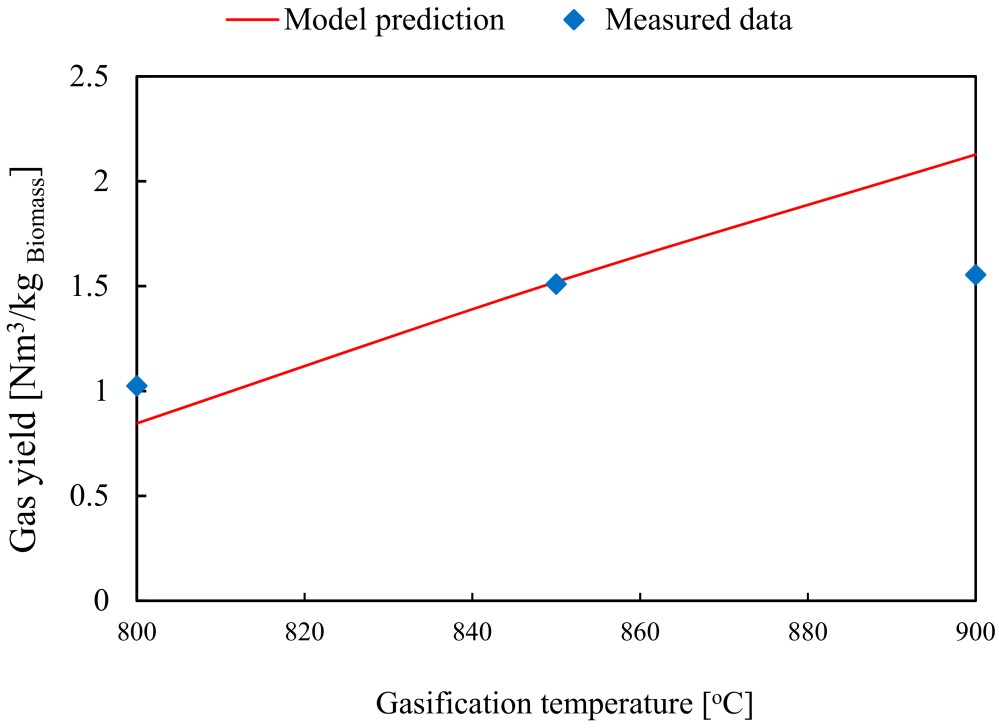

**Figure 5.** Effect of gasification temperature on the gas yield.

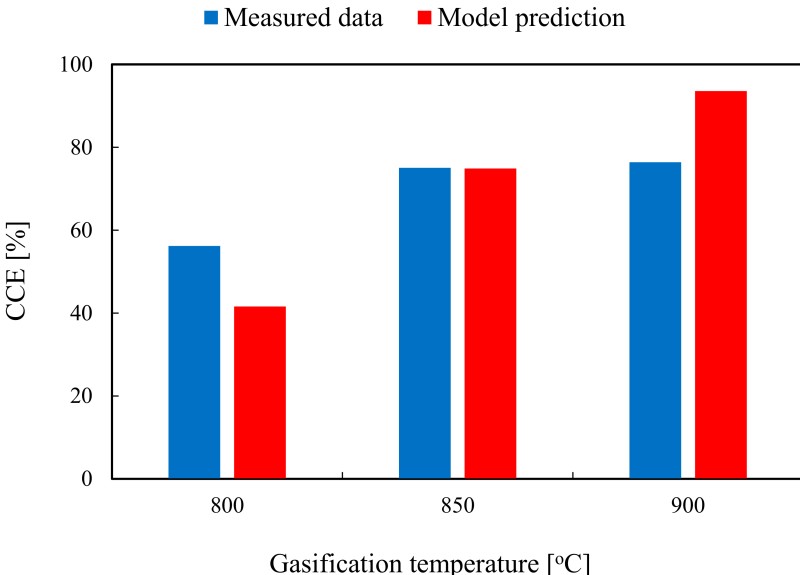

**Figure 6.** Effect of gasification temperature on carbon conversion efficiency.

Data analysis of the effect of gasification temperature is shown in Table 6. The mean errors of $H_2$ and $CH_4$ are the lowest and highest values, respectively. These errors are acceptable to predict the performance of the biomass gasification process. The differences between simulation and experimental results in the effect of gasification temperature are due to some simplified calculations and assumptions during the simulation. Biomass produces more tar and heavier hydrocarbons at lower temperatures, and they are decomposed at high temperatures. Tars released during biomass decomposition are assumed to be converted completely into CO and $CH_4$; therefore, the effect of temperature on tar production and decomposition is generally ignored in this study. Additionally, some reactions, i.e., steam reforming, char combustion, and hydrogen combustion, etc. are not considered in FORTRAN kinetic calculations, resulting in the high content of $H_2$ and $CH_4$ and the dry gas yield as well as the low fraction of $CO_2$ compared to experimental results. The equilibrium assumption replaces methane for all other hydrocarbons in the product gas and a negligible deviation of methane content between simulation and experimental results as observed in Figure 4.

**Table 6.** Analysis of data.

| Parameters | Mean Error | | | | | |
| --- | --- | --- | --- | --- | --- | --- |
| | $H_2$ | CO | $CO_2$ | $CH_4$ | Gas Yield | CCE |
| T (°C) | 0.115 | 0.17 | 0.222 | 0.303 | 0.235 | 0.2 |
| SBR | 0.193 | 0.174 | 0.329 | 0.134 | 0.076 | 0.033 |

### 3.2. Effect of Steam-to-Biomass Ratio

The steam-to-biomass ratio (SBR) represents a ratio of the mass flow rate of steam to biomass fed into the reactor [48]. Along with operating temperature, the steam-to-biomass ratio is a crucial operating parameter that affects significantly hydrogen production from biomass gasification [49]. The SBR was investigated in this study ranging from 0 to 1.6, and the temperature was at 850 °C.

In the following Figures 7–12, the simulation results were compared with experimental data for gas composition at various steam-to-biomass ratios. Generally, the $H_2$ and $CO_2$ fractions increase (see Figures 7 and 8), while CO and $CH_4$ show a downward trend (Figures 9 and 10). In the range of 0 to 0.9, the $H_2$ concentration rises considerably. Afterward, its increase slows down in the higher SBRs. A steady rise in the content of $CO_2$

is found in the SBR range from 0.7 to 1.6, reaching a peak of 16.1% at the SBR of 1.6. Those trends of the model predictions are similar to those found from the experimental data. Compared with other species, the difference between simulation results and experimental data in CH$_4$ content is the lowest, while in the case of CO$_2$, the fraction is the highest with the mean error of 0.329. The simulation results for hydrogen and carbon monoxide also display a good qualitative prediction of experimental data in the whole range of SBR. The mean errors of the effect of SBR on gas composition, presented in Table 6, are in acceptable ranges.

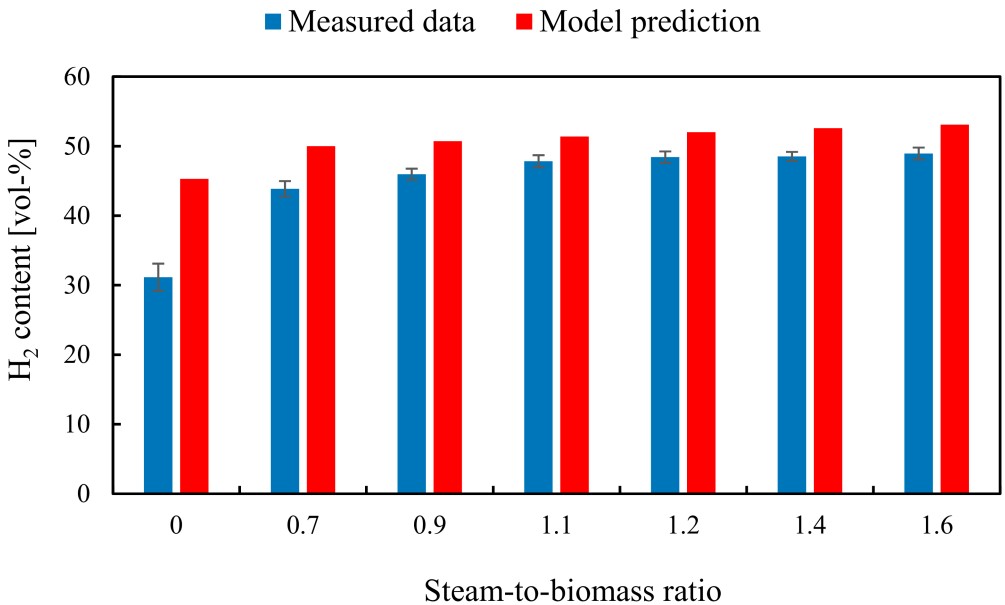

**Figure 7.** Effect of SBR on the hydrogen content.

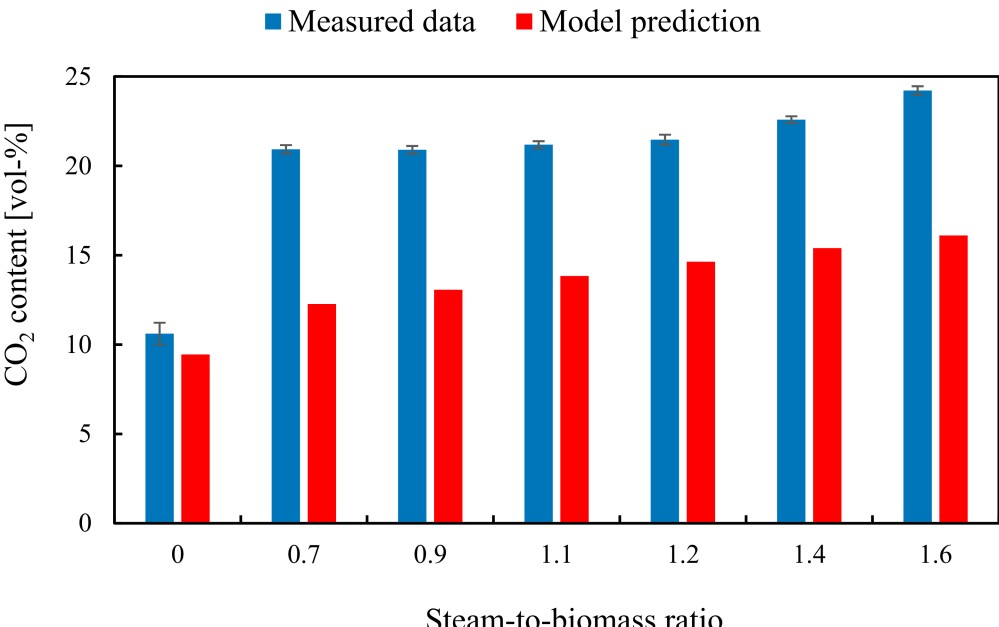

**Figure 8.** Effect of SBR on carbon dioxide content.

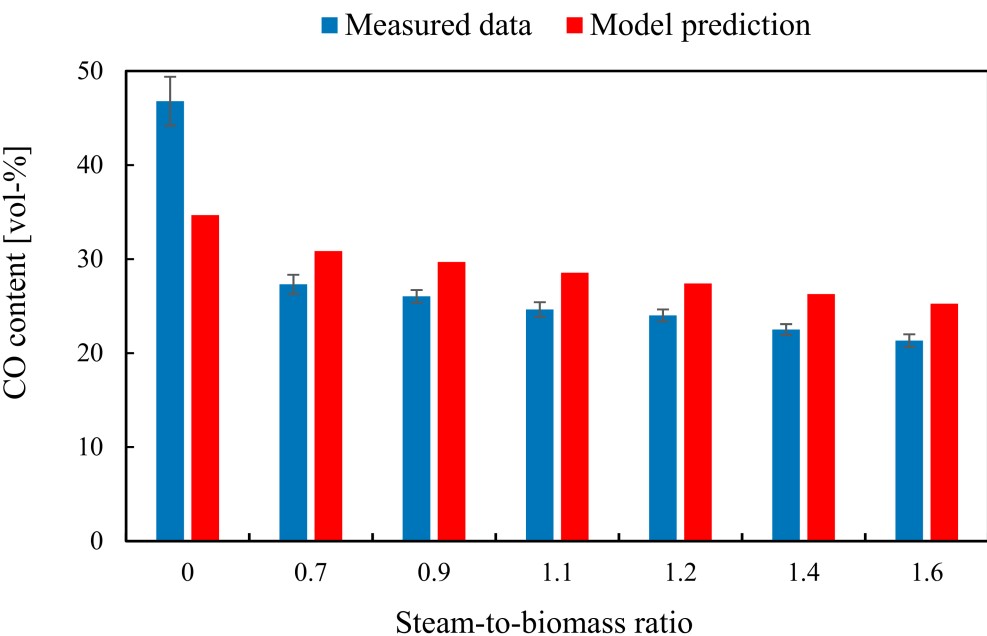

**Figure 9.** Effect of SBR on carbon monoxide content.

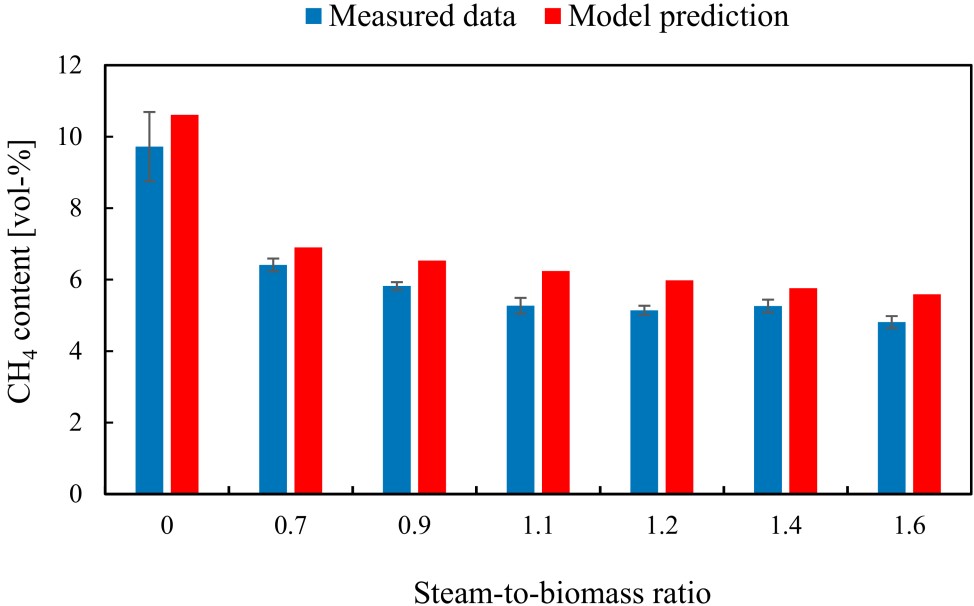

**Figure 10.** Effect of SBR on methane content.

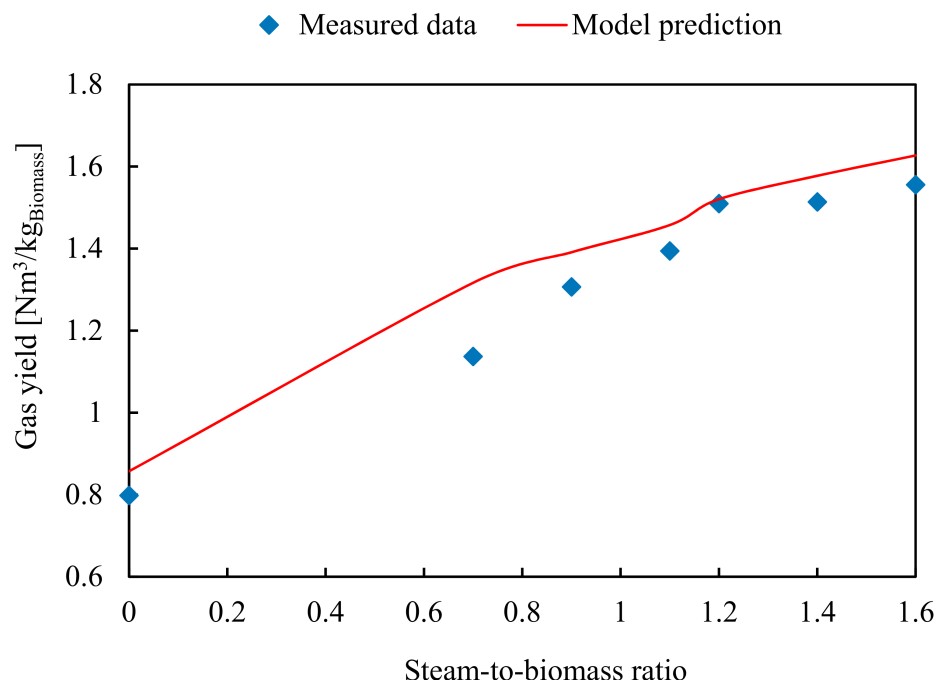

**Figure 11.** Effect of SBR on dry gas yield.

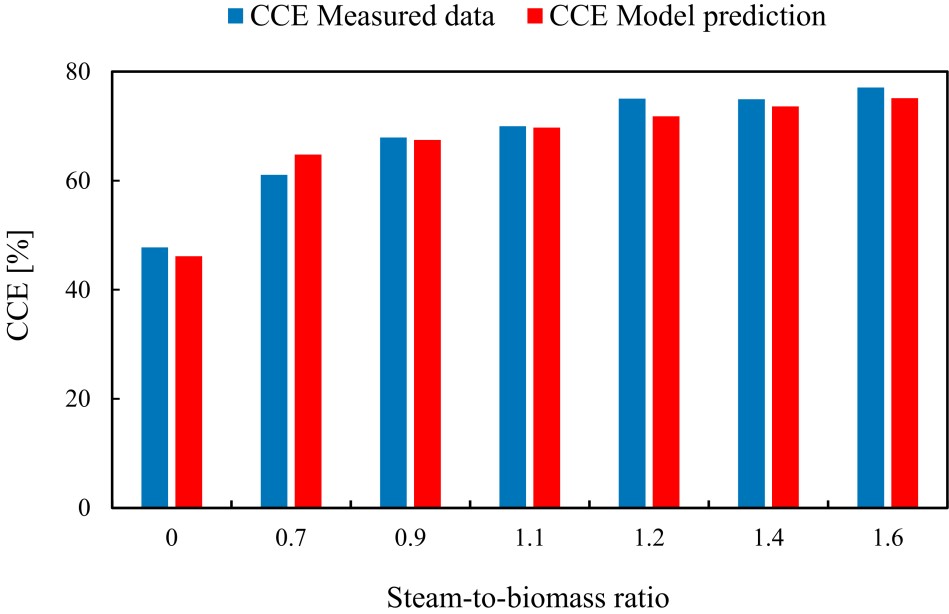

**Figure 12.** Effect of SBR on carbon conversion efficiency.

The dry gas yield of the product gas as a function of SBR is illustrated in Figure 11. As can be seen in the figure, the gas yield increases rapidly until an SBR of 0.9 before slowing down. It is noted that simulation results for gas yield are better in agreement with measured data from SBR of 1.2. Its mean error is about 0.17 that is acceptable for the investigation of steam-to-biomass ratios.

Figure 12 shows the comparison of the simulation results with the measured data for carbon conversion efficiency versus the SBR in the range of 0–1.6. Increasing trends of CCE are observed in both simulation and experimental results. A high SBR improves the gasification process and increases efficiency. The efficiency rises considerably at low SBRs, then its rate decelerates at high SBRs. The high accuracy of model predictions in carbon conversion efficiency compared to experimental data is determined by the mean error of 0.033.

The effect of SBR on the performance of biomass gasification is mainly due to the reactions with the presence of steam. The high amount of steam in the gasifier promotes char gasification and water–gas shift reactions, resulting in increasing hydrogen production and the amount of char consumed. The largest difference in $CO_2$ content is observed in this evaluation. This error is due to the combustion reactions of char and CO which are neglected. Additionally, this difference could be attributed to equilibrium assumptions. As discussed above, the main error of the simulation is likely attributed to simplified calculations and assumptions during the simulation. The ignorance of the effect of tar and some reactions causes the differences between simulation results and experimental data. In the presence of steam in the gasifier, the steam reforming reaction increases the decomposition of tar components at high temperatures. Generally, the model predictions and their error are capable of simulating the gasification performance under various steam-to-biomass ratios.

In summary, the study has developed a kinetic model which incorporates both reactor hydrodynamics and reaction rate kinetics to simulate biomass gasification in a bubbling fluidized bed reactor. The model is validated against experimental data at the same operating parameter range in terms of gasification temperature and steam-to-biomass ratio. It is noted that high agreement has been found with experimental results in most cases with the mean error ranging from approximately 0.033 to 0.329 (when fractions and yields of gas components and carbon conversion efficiency are considered). The trends observed in both simulation and experimental investigations are similar. There are slight deviations during the comparisons due to model assumptions. Additionally, the equilibrium model assumes that the reactions can reach a complete equilibrium, while the experimental conditions deviate from the ideal operating conditions, resulting in those errors. Finally, the model is capable of predicting the performance of biomass gasification in a bubbling fluidized bed and is sufficient to provide a good understanding of the phenomena occurring inside the gasifier.

## 4. Conclusions

In this work, an Aspen plus model for the steam gasification of biomass was developed to investigate the effect of operating parameters on the gasification process at steady-state conditions. Hydrodynamic and reaction kinetic calculations were implemented in external FORTRAN codes. Pyrolysis yield distribution obtained from the model of Neves was used to determine the mass yields of the decomposition of torrefied biomass. The simulation results for the product gas composition and carbon conversion efficiency versus gasification temperature and steam-to-biomass ratio were validated by the experimental data. The following relevant conclusions can be obtained from this study:

1.  At higher temperatures, the gasification process is favored. Here, the hydrogen production and the carbon conversion efficiency are increased, while the amount of carbon monoxide and methane in the product gas is decreased.
2.  Increasing the steam amount in the reactor promotes the performance of biomass gasification. The SB steam-to-biomass ratio strongly enhances the content and yield of hydrogen in the product gas as well as improves the gas yield and the carbon conversion efficiency.
3.  It is noteworthy that the model predictions are in good agreement with the experimental data, and the model is capable of simulating the performance of biomass gasification under various operating conditions, i.e., operating temperature and steam-to-biomass ratio. The minor deviations between the simulation model and the measured data are related to the model limitations, i.e., simplified calculations in bed hydrodynamics and kinetics.

The model-predicted values showed good agreement with experimental data. The errors of the simulation compared to experimental investigations could be attributed to process assumptions to simplify the calculations and the lack of tar decomposition reactions as well as the chemical equilibrium. The residence time in a fluidized bed gasifier might be not sufficient to reach equilibrium conditions in the experimental conditions. In further

studies, tar decomposition and combustion reactions are taken into consideration along with sufficient hydrodynamic calculations to reduce the deviations in the simulation of biomass gasification.

**Author Contributions:** N.M.N. is responsible for administration, conceptualization, the original draft, and the applied methodology. The simulation was conducted by N.M.N. F.A. supported the writing process with his reviews and edits. B.E. supervised the research progress and the presented work. All authors have read and agreed to the published version of the manuscript.

**Funding:** This research received no external funding. The corresponding author would like to thank the Technical University of Darmstadt, enabling the open-access publication of this paper.

**Institutional Review Board Statement:** Not applicable.

**Informed Consent Statement:** Not applicable.

**Data Availability Statement:** The data presented in this study are available in the article.

**Conflicts of Interest:** The authors declare no conflict of interest.

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
