# Peer review of "Process Simulation of Steam Gasification of Torrefied Woodchips in a Bubbling Fluidized Bed Reactor Using Aspen Plus"

_applsci, doi:10.3390/app11062877_

Round 1

Reviewer 1 Report

Dear Authors,

It is much more better than their previous version and they have tried hard to answer many comments and also changed a lot. I appreciate your work. Please find below comments on your work.

1- The abstract is still small and suffer from a complete abstracts. 

2- References are missed and mentioned with ERROR! due to the conversion or ....

3- Please check the dimension of reaction kinetics again! to make sure everything works perfectly. Are you sure about kPa-1 min-1? 

Author Response

Thank you very much for the evaluation of our manuscript. The comments and suggestions from the reviewers were very interesting, and the revision was performed accordingly. In the following, we provide specific responses and revisions to the reviewers’ comments.

Reviewer 2 Report

Line 267. Please change the number of figure.  Instead of Figure 1 should be Figure 3.

Line 309. Please add "Figure 5. Effect of gasification temperature on the gas yield" to the manuscript.

Line 343. Authors wrote, "In the following Figures...". Please add a number of figures.

Lines 49-59. Please change the font size.

Lines 135/139/214/287/292/315/329/343/345/370. Please remove "Error! Reference source not found.Error! Reference source not found.." and add a number of tables and figures.

I suggest  authors include some references about the kinetics of biomass gasification like e.g.(https://doi.org/10.3390/en11082126, https://doi.org/10.3390/en13174472)

Author Response

Thank you very much for the evaluation of our manuscript. The comments and suggestions from the reviewer were very interesting, and the revision was performed accordingly. In the following, we provide specific responses and revisions to the reviewers’ comments

Reviewer 3 Report

The use of models helps the development of new predictive methodologies to improve techniques. In the model I observe limitations due to the boundary conditions.

The units are wrongly indicated, as well as the line of multiplication symbols 182. 7.97.109 [m3 / mol] 0.5.s-1

in many lines the references are missing (135, 139, 292, 343 ...), the document is deficient in references.

Figure 1 is incomplete, units are missing. You need to make a better explanation of the figure.

There are two tables 1. The second of the tables has serious errors.

On line 208, the ecucacion is out of place.

It would be convenient to talk about the formation of TARS in gasification processes and its implications in the model.

Figure 5 is not in the text.

The references are not formatted correctly, they do not distinguish between first and last names.

The model overestimates the possibilities of generating gasification gas. In general, the document can be greatly improved since it has not been edited correctly. Improve document editing.

Author Response

Thank you very much for the evaluation of our manuscript. The comments and suggestions from the reviewer were very interesting, and the revision was performed accordingly. In the following, we provide specific revisions and responds to the reviewers’ comments. Please see the attachment.

Reviewer 4 Report

This paper has a large similarity index with the below paper. Therefore, I do not see a significant level of originality and novelty. 

''Nikoo, Mehrdokht B., and Nader Mahinpey. "Simulation of biomass gasification in fluidized bed reactor using ASPEN PLUS." Biomass and bioenergy 32, no. 12 (2008): 1245-1254.''

Author Response

Thank you very much for the evaluation of our manuscript. The comments and suggestions from the reviewer were very interesting, and the revision was performed accordingly. We would like to explain the improvement and differences between our study and the literatures. Please see the attachment.

Round 2

Reviewer 3 Report

Equation numbering order is not aligned.

The use of points as a sign of multiplication (example table 2) can lead to confusion.

Remove any paragraphs you don't use from the template.

The data in table 1 is not aligned.

Please check the references again, (for example) in the first reference, no authors appear.

Author Response

Dear reviewer #3,

We would like to thank you for the evaluation of our manuscript. The authors have made specific changes to improve this paper based on the valuable suggestions from the reviewer. We hope that you will find the updated manuscript satisfactory and acceptable for publication. Please see the attachment.

Reviewer 4 Report

Thank you for providing your feedback. 

Based on the provided explanation, I recommend accepting the paper (after making the below amendments)

Please describe the improvement and the differences of your work from Nikoo's paper in the revised manuscript (Explicitly and as provided in your notes, below).

''

  1. A combination of the Langmuir-Hinshelwood kinetic model and volumetric model was applied in this simulation for the reaction rates for char gasification with H2O and CO2. The kinetic parameters were obtained from our experimental investigations with the coefficient of determination R2 ranging approximately 86 to 90.6%. It is noted that Langmuir-Hinshelwood kinetic model is one of the most common kinetic expressions to explain the kinetics of the heterogeneous reactions, especially char gasification reactions.

The reaction model used in Nikoo et al. is the shrinking core model and Arrhenius model.

  1. Three chemical reactions, i.e. char gasification with steam and CO2 and water gas shift reaction, are taken into consideration in our model to calculate variations of species in biomass gasification. While Nikoo’s study mainly focused on two reaction rates, such as combustion and steam gasification.
  2. RSTOIC reaction blockwas used in our study to perform the reactor biomass gasification in a bubbling fluidized bed reactor, while RCSTR was implemented in the study in the literature.
  3. The mass yield distribution of pyrolytic products during biomass decomposition was derived from a pyrolytic model (Neves’s model) based on biomass’s properties. Due to the complexity of biomass characteristics, the determination of the mass yield distribution of pyrolytic products is crucial for the simulation of biomass gasification.

With the improvements in the simulation, the model predictions of this study are in good agreement with the experimental data. Some of the mean errors are better than these figures from the literature, e.g. the contents of hydrogen and methane. Therefore, this model can give better predictions of the performance of biomass gasification. 

''

Good Luck 

Author Response

Dear Reviewer #4

Thank you very much for the evaluation of our manuscript. The authors have made specific changes to improve this paper based on the valuable suggestions from the reviewer. We hope that you will find the updated manuscript satisfactory and acceptable for publication. Please see the attachment

This manuscript is a resubmission of an earlier submission. The following is a list of the peer review reports and author responses from that submission.

Round 1

Reviewer 1 Report

  1. In the abstract and conclusion section, some relevant and quantitative results should be anticipated.
  2. Why the authors choose woodchips cake instead of others? 
  3. What kind of elemental analyzer was used to determine the carbon, hydrogen, sulfur, and nitrogen contents, what type of apparatus was used to perform the proximate analysis, and what kind of mill use (type, company)?
  4. The results of model validation should be presented (described in chapter 2.5) 
  5. Please should add a reference about the steam gasification of biomass. I suggest the authors include some references like e.g. (https://doi.org/10.3390/en13174472, https://doi.org/10.1016/j.renene.2016.08.069)
  6. In table 1, steam concentration should be provided.
  7. Authors should compare the obtained results with those found in the literature. It is not sufficient to give values without explanation.
  8. Why a bubbling fluidized bed reactor was applied?

Author Response

Dear reviewer 1,

Thank you very much for the evaluation of our manuscript. The comments and suggestions from the reviewer were very interesting, and the revision was performed accordingly. In the following, we provide specific responses to the reviewers’ comments. We hope that you will find the updated manuscript satisfactory and acceptable for publication.

Reviewer 2 Report

Dear Authors,

Many thanks for your paper submission and it has very interesting data inside. However, there are many aspects which needs to be completely clarified. Please find below comments on your manuscript:

1- The general comment is that the paper is not in the format of Journal and must be arranged according to the journal requirements.

2- The title is very general and I can not see any specific case! There are plenty works very similar to this title.

3- The abstract sentences are quite well-known for all people in this topic! We know these introduction inside abstract and please be more specific! 

4- You mentioned inside the abstract that: The results are in a very good agreement with experiments! Please be more specific! How close they are? 

5- The novelty is not mentioned inside the introduction! Please clearly describe at the last paragraph what you have added to this topic! 

6- I can not see the clear description and the shape of the experimental facilities! The capacity, the sand depth and etc. 

7- Where is this biomass came from please mention the origin of it! Which kind of biomass is that? We need to see clear data! 

8- I can not see that the experiments and simulation data are close! Please provide the error bars! 

9- Where is your uncertainty analysis to help us the error sources in simulation and experiments! 

10- Too many special assumptions you have! Please describe that why you consider some of them as inert! Please provide some references that did the same! 

11- What is the average porosity of your biomass. 

Author Response

Dear reviewer

Thank you very much for the evaluation of our manuscript. The comments and suggestions from the reviewer were very interesting, and the revision was performed accordingly. In the following, we provide specific responses to the reviewers’ comments. We hope that you will find the updated manuscript satisfactory and acceptable for publication.

Reviewer 3 Report

Comments:

The authors have conducted a numerical investigation on steam gasification of torrefied woodchips in a bubbling fluidized bed reactor by adopting Aspen Plus simulator. Unfortunately, this article is suffering from significant drawbacks, and some of them are listed below:

  1. In general, an ‘abstract’ of a reader-friendly scientific article is a ‘self-standing summary’ of the whole investigation, which usually includes the definition of the ‘problem,’ the ‘brief description of the adopted methodologies, and ‘important research findings.’ The authors have tried to organize their abstract as mentioned above!! However, unfortunately, the current version of this ‘abstract’ does not satisfy the above conditions. For example, the authors have used their experimental data for validation purposes; it is not mentioned explicitly. Besides, they have not strongly highlighted their important findings in this abstract.     
  2. The next important part of an article is the ‘introduction.’ It is generally treated as the ‘brain’ of an article. Consequently, the ‘introduction part’ controls/guides the flow of the construction of the rest of the article’s parts. So, a question arises on how to construct the ‘introduction’ of an article? It is a significant event for a well-organized scientific article. To address the above question, it is required to survey the existing literature on the subject matter of the article extensively, and this kind of literature survey will help authors to reveal a ‘research gap (=> this can be termed as the ORIGINALITY of the research article)’ within the existing literature. Once the ‘research gap’ is identified, then the rest of the article MUST be devoted to filling up the ‘research gap’ as determined. This kind of construction of the ‘introduction’ of an article enhances the understandability of the potential readers, even those who are not experts on the topic of the article’s investigation.
  3. It is noticed after a specific level of discussion that the authors have stated that ‘the number of simulation studies on biomass gasification is still limited’. Then, the authors have jumped suddenly into their study by saying that ‘In this work, a process simulation model for biomass gasification was developed in Aspen Plus.’ Since ‘Aspen Plus’ is a commercial software, I believe that literature related to this software’s utilization is sufficient!  However, it seems that the authors have tried to write down their ‘introduction’ as stated above!! Unfortunately, the current version of this article’s ‘introduction’ has not been written by conducting a continuous and systematic survey of the existing literature in an extensive manner as described above. Therefore, the current version of ‘introduction’ MUST be revised appropriately as highlighted above; this event is a mandatory event for this article.  
  4. Since the ‘research gap’ has not been revealed systematically, the other parts of this article has not been placed in the right order. Consequently, the potential readers will encounter difficulties while going through this article. Most importantly, the general ‘norm’ of the numerical investigation has not been maintained in this article!! For example, the authors have stated that ‘a simulation model for biomass gasification was developed in Aspen Plus’. Unfortunately, the adopted numerical procedures have not been presented in this article systematically. For instance, governing equations, numerical domain/physical domain (test rig), numerical conditions, boundary conditions, and so forth are missing in this article. Besides, it is essential to describe in the first place what is the default version of ‘Aspen Plus’? What is the capability of ‘Aspen Plus’? Then, they have to write down the limitations of ‘Aspen Plus’ regarding the author’s research target. Consequently, it is necessary to modify it by writing an external FORTRAN code to achieve the author’s target precisely by adopting ‘Aspen Plus’ simulator. Therefore, it is strongly recommended to revise the ‘numerical part’ precisely as it is the main event of this article.
  5. The significant deviation exists in all comparisons (between numerical- and experimental-results) presented in this article, and the causes of these deviations have not been addressed explicitly. Although some implicit explanations are found regarding this matter in this article, those explanations are not enough!! Besides, how these deviations (significant deviations are seen in Figs. 2, 3 & 9 for instance, and other Figs. as well) can be reduced?   
  6. The ‘results and discussion’ section MUST be enriched and enlarged by selecting much more appropriate data and corresponding discussions based on the ‘research gap’ as determined in the part of ‘introduction’. To do so, the authors have to pay their attention very seriously and carefully. This is also a mandatory event for this article.                                                                                                       
  7. This article has many other problematic events that are not listed here. It is recommended to find out those events by themselves (authors) using repeated and careful examinations, and afterward, revise them accordingly along with the above-stated shortcomings in such a manner so that the potential readers can easily understand this article without any potential difficulty. Anyway, please wait for the comments from the ‘editorial office’ of ‘Applied Sciences’.

Author Response

(The authors gave the same response as above.)

Round 2

Reviewer 2 Report

Dear Authors,

Many thanks for the revised version. Your answers were quite good and reasonable. I am satisfy about what you did. Please be really careful for the conversions! I still can see that the abstract and references are still messy. You have two reference lists! one 39 refs. and other 38 refs. Please remove one of them and check that all references have been properly cited. 

Reviewer 3 Report

Please wait for the comments from the editorial office of Applied Sciences.